# The Use of Apple Vinegar from Natural Fermentation in the Technology Production of Raw-Ripened Wild Boar Loins

**DOI:** 10.3390/foods12213975

**Published:** 2023-10-30

**Authors:** Anna Łepecka, Piotr Szymański, Anna Okoń, Beata Łaszkiewicz, Sylwia Onacik-Gür, Dorota Zielińska, Zbigniew J. Dolatowski

**Affiliations:** 1Department of Meat and Fat Technology, Prof. Waclaw Dabrowski Institute of Agriculture and Food Biotechnology—State Research Institute, 02-532 Warsaw, Poland; piotr.szymanski@ibprs.pl (P.S.); anna.okon@ibprs.pl (A.O.); beata.laszkiewicz@ibprs.pl (B.Ł.); sylwia.onacik-gur@ibprs.pl (S.O.-G.); zbigniew.dolatowski@ibprs.pl (Z.J.D.); 2Department of Food Gastronomy and Food Hygiene, Institute of Human Nutrition Sciences, Warsaw University of Life Sciences-SGGW, 02-776 Warsaw, Poland; dorota_zielinska@sggw.edu.pl

**Keywords:** game meat, wild boar, quality, acetic bacteria, *Sus scrofa*

## Abstract

Wild boar meat is difficult to process, mainly due to its hardness and stringiness. Three types of raw-ripened wild boar loins were produced (C—control treatment, R1 and R2—treatments with the addition of apple vinegar in various production variants). The research aimed to develop a new innovative technology for the production of wild boar loin using apple vinegar for marinating and to determine the impact of apple vinegar on the microbiological and sensory quality, and physico-chemical parameters of the product. As part of the research, a technology for the production of ripened wild boar products was developed and the composition of fatty acids, cholesterol content, pH value, oxidation-reduction potential, thiobarbituric acid reactive substances (TBARS) index, color, microbiological, sensory, and statistical analysis were determined. It was found that the loins were characterized by a high content of saturated, monounsaturated, and polyunsaturated fatty acids (20.18–43.37%), a low content of trans fatty acids (0.30–0.57%), and a high cholesterol content (75.13–85.28 mg/100 g of the product). Samples with apple vinegar (R1 and R2) were characterized by a lower pH value (5.10–5.70; *p* < 0.05), a comparable oxidation-reduction potential (409.75–498.57 mV), and a low TBARS index (0.461–1.294 mg malondialdehyde/kg of product). Their color was lighter (L* 38.25–40.65). All the tested loins were characterized by appropriate microbiological quality guaranteeing the storage durability of the product. R1 and R2 treatments were characterized by the greatest juiciness. The highest overall quality was achieved by R1 loins (7.36–7.76 c.u.). The apple vinegar used to marinate the loins had a positive effect on their microbiological and sensory quality as well as physico-chemical parameters. Moreover, the technology guarantees the appropriate quality and health safety of the products.

## 1. Introduction

Concerning climate change, which the whole world is dealing with, it is important to limit the negative impact of man and the economy on the natural environment. To reduce the production and consumption of meat from farmed animals, the consumption of wild animal meat becomes an alternative. Following the principle of sustainable development of forest areas, the goal of forestry is to adapt the number of wild animals to the changing environment [1]. The lack of human interference in the way of feeding wild animals means that the meat obtained from game animals is considered organic. Wild meat meets ethical standards for animal welfare, and farming in the wild has less impact on environmental ecosystems compared with factory farming [2].

Game meat consumption in Europe is low (usually less than 1 kg/person/year) and largely varies by country and region. In regions with a strong hunting tradition, game consumption is higher [3]. It is estimated that only 2–4% of the European population regularly eats this type of meat. [4]. The main reason for the low consumption of game meat in Europe is its high price, availability, lack of culinary traditions, and concerns about proper slaughter hygiene [3]. An increase in the population of wild game and the need to reduce its number related to African swine fever (ASF), has resulted in an increased supply of venison and there is a need to manage it [5]. This may translate into a higher consumption of game meat in the world in the future.

The interest in venison among consumers is constantly growing, mainly due to its nutritional and health-promoting qualities [6,7]. Wild animal meat is a food rich in vitamins and nutrients and low in fat, with the right ratio of polyunsaturated fatty acids (PUFA) to saturated fatty acids (SFA) and the right ratio of n-6/n-3 fatty acids [2,8]. The main food for wild animals is nature, and not, as in the case of farmed animals, feed, which is often based on GMOs. The use of natural food by the game allows it to absorb vitamins and nutrients straight from the source. Additionally, game animals must find their food; therefore, the animal is constantly on the move. It is a natural raw material without any negative features associated with the intensive, industrial breeding of slaughtered animals. The growth rate of game animals is not accelerated, and free access to a variety of food affects the specific taste and unique properties of this raw material. Wild animals are not exposed to the stress associated with industrial farming, and properly hunted animals do not experience the stress associated with the journey to the slaughterhouse, and the meat obtained from them contains only trace amounts of adrenaline. Wild animal species are monogastric. Therefore, the assumption of fatty acids is directly related to concentration in muscle and fat tissue [9]. Compared with pork, wild boar meat has a lower caloric value, and lower fat content, and contains more vitamins D, E, B_2_, B_12_, and folates as well as potassium, iron, zinc, manganese, and iodine [10]. Culinary game meat is more tender, characterized by high water absorption and low weight loss during storage [11]. The lack of the use of chemical additives, hormones, or antibiotics also speaks in favor of venison [12].

Wild boar meat is difficult to process, mainly due to its hardness and stringiness, resulting primarily from the characteristics of the connective tissue and the activity of proteolytic enzymes during carcass cooling. Ripe meat is characterized by greater tenderness and a better taste and aroma. Therefore, following the correct rules during post-slaughter treatments is crucial and affects the quality of meat [8]. In addition, the durability of the finished product is largely influenced by the microbial quality of the raw material.

One way to improve the culinary quality of wild boar meat is marinating. Marinades consist of a spice mixture that includes vinegar, salt, oil, and spices. The marinating process involves placing the meat for several days and is intended to soften the muscle tissue by increasing the degree of proteolysis of the meat. This happens by lowering the pH of the meat, which stimulates the enzymatic activity of proteolysis during meat maturation. The addition of organic acids, such as citric, acetic, or tartaric acid, speeds up the maturation time of the meat. The mechanisms of action of organic acids include weakening structures during meat swelling, increasing proteolysis thanks to cathepsins, or increasing the conversion of collagen to gelatin, which occurs at low pH [13].

Apple vinegar is not a new product; it has been used in meat marinating technology for hundreds of years. It has been shown to have a positive effect on the sensory characteristics of meat, improve palatability, increase tenderness, improve meat color, guarantee microbiological safety, and enrich it with active ingredients [14]. However, it should be emphasized that, so far, vinegar has been used to improve the culinary properties of uncured meat, and its industrial use in the case of cured meat has been limited.

Due to the specific properties of wild boar meat, it is required to use the appropriate processing technology to obtain meat products of suitable desirability. To improve the quality of raw meat, IBPRS-PIB (Instytut Biotechnologii Przemysłu Rolno-Spożywczego—Państwowy Instytut Badawczy) in Poland attempted to marinate the raw material in apple vinegar, derived from natural fermentation. As part of the “Polish Fruit Vinegar” project at IBPRS-PIB, an innovative fruit vinegar product using local strains of microorganisms (bacteria and yeast) was developed and implemented into production [14]. The result was a biodiverse product with health-promoting properties, free of chemical preservatives, containing natural metabolites of microorganisms, with high storage stability and repeatable sensory properties [15]. Moreover, the study by Łepecka et al. [16] describes the use of apple vinegar as a marinade for pork. It was found that the marinade had a positive effect on the meat products, increasing their storage stability, and obtaining the appropriate microbiological and sensory quality.

It was hypothesized that the addition of apple vinegar to the marinating process of wild boar loins would have a positive effect on their biochemical and microbiological evaluation and would improve the physical and chemical properties of the finished product. This study aimed to apply a technology for the production of raw-ripened wild boar loin with the use of apple vinegar in the process of marinating the raw material and to determine the effect of the addition of apple vinegar on the microbiological and sensory quality as well as the physical and chemical parameters of the finished product.

## 2. Materials and Methods

### 2.1. Materials

#### 2.1.1. Apple Vinegar

An innovative technology for the production of apple vinegar was developed at IBPRS-PIB as part of the “Polish Fruit Vinegar” project [14]. The raw materials for the production of apple vinegar were cold-pressed, unpasteurized apple juices (Champion), which were first anaerobically fermented (at 25 °C) with the Tokay wine yeast. Next, the components of acetic biosynthesis (at 30 °C) were conducted with cultures from the IBPRS-PIB collection (*Acetobacter pasteurianus* O4 (KKP 674; GenBank OM200034) and *Acetobacter pasteurianus* MW3 (KKP 2997; GenBank OM212983) [14,15]. The characterization of raw materials was carried out in the study by Gajewska et al. [17]. The number of yeasts and molds was <1.00 log CFU/mL, the total extract content was 11.4–15.1 g/100 cm^3^, the total sugar content was 10.5–14.7 g/100 cm^3^, the total acidity was 0.32–0.56 g/100 cm^3^, and the pH value was 3.45–3.93. The heavy metals, cadmium and lead, were not detected (values below the limit of quantification, <0.003 mg/kg and 0.020 mg/kg, respectively). Then, the wines made after the first stage of fermentation were also tested and the total sugar content (<1.0 g/cm^3^) and alcohol content (6.81–10.84%) were determined. In the biosynthesis process, the final product was obtained, which was apple vinegar with an alcohol content below 1%, and vinegar strength of 3.3–4.5 g of acetic acid/100 cm^3^ [17]. The calcium content was 64.4 mg/L, manganese 1.67 mg/L, magnesium 60.3 mg/L, zinc 0.541 mg/L, copper 0.480 mg/L, iron 0.805 mg/L, potassium 1.197 mg/L, sodium 11 mg/L, and NaCl 0.0027 mg/L. No presence of heavy metals cadmium, lead, or mercury was detected (values below the limit of determination <0.003 mg/L, <0.020 mg/L, <0.001 mg/L, respectively) [18]. The pH value of apple vinegar was approximately 3.00, the total acidity was 0.4–0.6 g of citric acid/L, the number of acetic bacteria was 1.4 × 10^6^ CFU/mL, and the vitamin C content was 0.72–0.95 mg/100 mL [17,18].

#### 2.1.2. Raw-Ripened Wild Boar Loins

Three independent production batches of raw-aged wild boar smoked meat were produced in industrial conditions. The meat was purchased from a meat processing plant that purchases and processes game meat. The plant is located in south-eastern Poland (Podkarpackie Voivodeship). The raw material was wild boar loin (*longissimus dorsi*), without fat and skin, in the amount of 200 kg per batch. In the first stage of the Incubator of Innovation 4.0 project [19], the authors prepared pilot wild boar products with various additions of apple vinegar (control sample without apple vinegar, 2, 3, 4, and 5% of apple vinegar added, respectively). Based on the physico-chemical, microbiological, and sensory tests carried out, a 4% addition of apple vinegar was selected for the production of wild boar loins. Three treatments for the loins were prepared:

Treatment C (control)—the raw material was manually mixed with curing salt (99.5% NaCl, 0.5% NaNO_2_; 1.6% of the weight of meat) and left to cure for 48 h at 4 °C. Glucose (5 g/kg of meat) and water (4% by weight of meat) were then applied to the meat and massaged by hand, after which the loins were suspended on smoking sticks;

Treatment R1—the raw material was manually mixed with curing salt (99.5% NaCl, 0.5% NaNO_2_; 1.6% of the weight of meat) and left to cure for 48 h at 4 °C. Glucose (5 g/kg of meat) and apple vinegar (4% by weight of meat) were then applied to the meat and massaged by hand, after which the loins were suspended on smoking sticks;

Treatment R2—the raw material was mixed manually with curing salt (99.5% NaCl, 0.5% NaNO_2_; 1.6% to the weight of meat), then apple vinegar was massaged into the raw material (4% to the weight of meat) and left to mature for 48 h at 4 °C, then glucose (5 g/kg of meat) was added to the meat, after which the loins were suspended on smoking sticks;

After the initial drying of the surface of the product, the maturing process was carried out at 15–17 °C and humidity of 75–80%. After 2–3 days, the products were smoked with cold smoke at 20–25 °C for 1–1.5 h. Then, maturing was continued for 4 weeks with the assumed initial parameters of the process and low air movement. With a weight loss of about 35%, the products were vacuum-packed and placed in a cooling room (4 °C). Samples for testing were taken at time 0 (immediately after production) and during 14 and 28 days of storage (time 1 and 2). The produced wild boar loins contained on average 55.4% water, 33.4% protein, 3.2 fat, and 2.79% NaCl. The content of water, protein, and fat, as well as the salt content, were determined based on the applicable ISO standards [20,21,22,23].

### 2.2. Methods

#### 2.2.1. Fatty Acid Composition and Cholesterol Content

The fatty acid (FA) profile was determined according to ISO 12966-1:2014 [24] using gas chromatography with a flame ion detector and a highly polarized column with BPX 70 phase (Hewlett-Packard 6890 II-FID; Agilent Technologies, Santa Clara, CA, USA). Individual fatty acids were related by comparing the retention times with a FAME standard mixture and expressed as a proportion of the sample. The results are presented as the sum of saturated fatty acids (SFA), the sum of monounsaturated fatty acids (MUFA), the sum of polyunsaturated fatty acids (PUFA), the sum of n-3 fatty acids, the sum of n-6 fatty acids, the n-6/n-3 ratio, and the sum of trans fatty acids.

Cholesterol content was analyzed using gas chromatography with FID—flame ionization detector (Hewlett-Packard 6890 II-FID; Agilent Technologies, Santa Clara, CA, USA) according to the IBPRS-PIB laboratory’s procedure [25].

#### 2.2.2. Determination of the pH Value

The pH value was determined using a FiveEasy F20 pH-meter with a LE438 electrode (Mettler-Toledo GmbH, Greifensee, Switzerland) according to ISO 2917:1999 [26].

#### 2.2.3. Determination of Oxidation-Reduction Potential (ORP)

The red-ox potential was determined with the SevenCompactTM S220 meter with the InLab Redox electrode (Mettler-Toledo GmbH, Greifensee, Switzerland) [27].

#### 2.2.4. Thiobarbituric Acid Reactive Substances (TBARS) Index

Lipid oxidation was performed by evaluating the TBARS index according to Pikul et al. study [28]. The color intensity of the reaction of malondialdehyde (MDA) with 2-thiobarbituric acid was measured using a spectrophotometer (U-2900 spectrophotometer; Hitachi, Tokyo, Japan) at a wavelength of 532 nm. TBARS values are expressed in mg of malondialdehyde (MDA) per kg of product.

#### 2.2.5. Instrumental Measurement of Color

The measurements of the color in the CIELab system (L* (lightness), a* (redness: green to red), b* (yellowness: blue to yellow) were taken using a CR-300 Chroma Meter (Konica Minolta, Tokyo, Japan). The parameters of the measurements were as follows: device Konica Minolta, CIE standard illuminant D65, observation angle 2°, no readings per sample 20, aperture size 8 mm [29]. The calibrations were conducted using a white standard L* 99.18, a* −0.07, b* −0.05.

#### 2.2.6. Microbiological Analysis

To determine the total viable counts (TVC), the nutrient agar (Biomaxima, Lublin, Poland) was used according to ISO 4833-1:2013 [30]. The lactic acid bacteria (LAB) were indicated on MRS agar (Becton Dickinson Polska Sp. z o.o., Warszawa, Poland) according to ISO 15214:1998 [31]. The counts of the *Enterobacteriaceae* bacteria (ENT) were enumerated on MacConkey agar (Biomaxima, Lublin, Poland) according to ISO 21528-2:2017 [32]. *Escherichia coli* (EC) counts were determined on *E. coli* Chromogenic Medium (Biomaxima, Lublin, Poland) according to ISO 16649-1:2018 [33]. The coagulase-positive staphylococci (*Staphylococcus aureus* and other species) (SA) were determined on Baird-Parker agar supplemented with egg yolk tellurite (Oxoid, Basingstoke, UK) based on ISO 6888-1:2021 [34]. The presence of *Salmonella* spp. (SAL) and *Listeria* spp. (LIST) was determined on XLD agar and ALOA agar, respectively [35,36]. The presence of *Campylobacter* spp. (CAMP) was marked on CCDA LAB-AGAR™ (Biomaxima, Lublin, Poland) according to ISO 10272-1:2017 [37].

#### 2.2.7. Sensory Analysis

The sensory profile of the loins was determined using the Quantitative Descriptive Profile (QDP) method [38]. The 2 mm thick slices of wild boar loin were placed in disposable, coded, plastic containers with a lid. A qualified 10-person team [39] from WULS-SGGW evaluated the loins in terms of odor (7 discriminants), color (1 discriminant), juiciness (1 discriminant), tenderness (1 discriminant), flavor (7 discriminants), and overall quality (1 discriminant), placing the rating on a scale of 0–10 c.u. Odor was assessed: 0—imperceptible odor, 10—very intense odor; color was assessed: 0—dark beige, 10—red; juiciness was assessed: 0—dry, 10—very juicy; fragility was assessed: 0—slightly fragile, 10—very fragile; flavor was assessed: 0—imperceptible taste, 10—very intense taste.

#### 2.2.8. Statistical Analysis

As part of the research, three independent production batches were made. Measurements were repeated at least three times for each production batch. The chemical composition tests were carried out at time 0 (after production), fatty acid composition at times 0 and 2 (after production and after 28 days of storage), and all other tests at times 0, 1, and 2 (after production, after 14 days and 28 days of storage). The obtained results are presented as mean and standard deviation (SD). To analyze the effects, a one-way analysis of variance (ANOVA) was performed and a significance level of *p* < 0.05 was established. Tukey’s HSD post hoc test was used to compare the mean pairs. For the calculations, the Statistica 13 software (TIBCO Software Inc., Palo Alto, CA, USA) was used.

## 3. Results and Discussion

Wild boar meat was characterized by a high content of unsaturated fatty acids (Table 1). The analyzed products had 20.18% to 34.78% of polyunsaturated fatty acids (PUFA), 27.40% to 43.37% of monounsaturated fatty acids (MUFA), and 31.07% to 38.67% of saturated fatty acids (SFA). In comparison, pork loin has approximately 6.46–15.21% PUFA, 40.96–52.62% MUFA, and 40.89–40.04% SFA [40,41]. The World Health Organization (WHO) recommends limiting the consumption of SFA to 10% of total energy intake and trans fatty acids (TFA) to 1% and replacing them with PUFA. The reduction in SFA and TFA in the diet decreases the risk of cardiovascular diseases by lowering LDL cholesterol and all-cause mortality [42]. Moreover, not only the amount of PUFA is important but also the n-6 to n-3 fatty acids ratio. Nutritional recommendations are that the proportions of those fatty acids should be less than 4:1. In the modern-day diet, the ratio is 10:1 and people have a deficiency of n-3 fatty acids [43]. In the analyzed products, this ratio was from 2.11 to 6.77, which indicates that half of the products fit these recommendations. Wild boar meat could be promoted for its higher nutritional value in comparison with other meats.

According to the literature, wild boar loin meat contains approximately 34.4 mg of cholesterol/100 g of product [44] and pork loin 84.76 mg/100 g of product [45]. Levels of this compound in the analyzed products were 75.13–85.28 mg/100 g of product. The higher cholesterol content could be the effect of the meat preparation method. During the process of maturing and smoking wild boar loins, the weight loss was 35%, which leads to an increase in its ratio in the meat product. During the maturation and smoking of wild boar loins, a decrease in meat weight was observed, which is caused by a loss of water and an increase in dry matter [46]. Therefore, the total cholesterol content may be higher than that reported in the literature. According to Smagowska et al. [47], cholesterol levels are usually higher in dried products compared with conventional products. Moreover, raw wild boar products are rich in saturated fatty acids, which contributes to an increase in cholesterol content [48]. Differences in fatty acids and cholesterol content among the samples and time of storage were statistically significant (*p* < 0.05). However, these results do not indicate a clear trend in the influence of apple vinegar, preparation, or storage method on these parameters. In this study, wild boar loins were used. Samples were coming from different animals. It is known that the gender, age, and karyotype of animals [44,49], as well as diet and season of the year [50], may influence the fatty acids composition and cholesterol content of animal lipids [2]. In the study by Vargas-Ramella et al. [46], an increase in monounsaturated fatty acids (MUFA) was observed in dried deer loins during the ripening process. The release of fatty acids is dependent on muscle enzyme systems, mainly lipolytic enzymes. Additionally, the release of free fatty acids (FFA) is observed during the ripening and storage of these products, which may also influence differences in the proportions of fatty acids.

The addition of apple vinegar affected the physicochemical parameters of dry fermented wild boar loins (pH, ORP, and TBARS). Table 2 shows the physicochemical properties of wild boar loins. The loins samples were characterized by a pH value in the range of 5.10–5.70, and an ORP index in the range of 409.75–498.57, while the TBARS values were 0.461–1.294 mg MDA/kg of product. One of the most important parameters influencing the quality of meat products is the pH value. From the first day of storage (4 °C), there was a significantly (*p* < 0.05) lower pH value in the sample R2 compared with the other treatments. After 14 and 28 days of storage, a decrease in the pH value was observed in sample R1 (0.12 and 0.14, respectively). The use of vinegar for marinating meat causes a decrease in the pH of the samples, which results from the denaturation of proteins by the acetic acid contained in the vinegar on the surface of the meat samples. On the other hand, the higher pH values may be attributed to the production of alkaline substances caused by proteolysis resulting from the growth of microorganisms during the manufacturing process [51]. Proteolysis is carried out by endogenous enzymes contained in meat and enzymes of microbial origin. Small peptides and free amino acids are then formed. Intensive proteolysis, followed by the degradation and decarboxylation of peptides, also promotes the production of biogenic amines. In the assessment of stability, the acidity in the R2 sample may be the result of the activation of proteolytic enzymes—cathepsins and calpains—and the increase in free amino acids and peptides (low pH at 0 time) which stabilize the pH value, while in other samples, the acidity increases (reduction in the pH value), which is also confirmed by our research. The low pH values of the vinegar varieties used in marination and the fact that the meat absorbs a certain amount of vinegar after marination are thought to be the reasons for the decrease in the pH value of meat products compared with the control group samples. Researchers have found that acidic marinades induce marinade absorption between muscle fibers, which accelerates muscle fiber swelling [52]. On the other hand, the marination process with organic acids causes may negatively affect the color and taste value of the meat [53,54]. According to Jones et al. [55], adding vinegar to beef meat products reduced the pH value from 5.64 to 4.91.

The ORP index can be used to assess the balance between the reactive oxygen species (ROS) formation and the degree of their specific neutralization. The addition of apple vinegar to meat maturation did significantly affect the ORP (mV) of the samples after production (*p* < 0.05) (Table 2). The ORP index of the samples increased with an increase in the storage period and these changes were statistically significant (*p* < 0.05) (except for the R2 test, where a significant systematic increase in the ORP index was observed only up to 7 days of storage). Taking into account the influence of the addition of apple vinegar on the ORP index, in the first research period (0 days), the R2 samples showed the best (lowest) ORP index, although, after 14 days, the index C tests showed the lowest ORP index. After 28 days, no significant statistical differences were found between the loins groups. The ORP index of meat products depends on many factors, including the nature of the conjugated redox pairs, the concentration of pro-oxidants, the concentration of antioxidants, and temperature [56]. The value of the redox potential can be related to the chemical changes taking place in the product; on the one hand the system stabilizes, on the other hand it is related to the concentration of components capable of accepting or donating electrons. Stabilization of the system was also observed in our research.

The TBARS values are expressed in mg of malondialdehyde (MDA)/kg of product; the values indicate the amount of the secondary oxidation product of fat. Secondary oxidation products are formed during the next stage of autoxidation, when peroxides are oxidized to aldehydes and ketones. The determination of the malondialdehyde content is one of the most commonly used methods for evaluating the degree of lipid rancidity [57]. The marinating method significantly (*p* < 0.05) affects the TBARS value. As presented in Table 2, after production, there was a significantly (*p* < 0.05) lower MDA content in the R2 sample. After 14 days of storage, the highest TBARS value was observed in C and the R2 tests and the lowest in R1. After 28 days of storage, there was a significantly (*p* < 0.05) higher MDA content in the R2 sample, which was twice as high as the other variants. A decrease in the MDA content during storage is because the secondary lipid oxidation products could be further degraded to other, more stable, compounds [58]. In our research, a similar trend was noted for the control treatment. Furthermore, during storage, the permitted quantity did not exceed the amount of MDA in meat products which is around 2–2.5 mg/kg of product [58]. The values shown in the table indicate that in the C and R1 loins there may be a stage of formation of so-called secondary oxidation compounds, while in the R2 test, the primary formation of compounds reacting with thiobarbituric acid occurs.

Wild boar meat has more intense red coloration than pig meat. These differences are related to the slower and less extensive decline in pH value and to the faster decline in temperature, which can be explained by the genetic group, management, and feeding of wild boars, resulting in older and less heavy animals at slaughter age [59]. Similar results were reported by Schwaegele et al. [60] and Hedrick et al. [61]. According to their results, wild animals have darker muscles than domestic animals due to a higher concentration of myoglobin as a result of their more intense physical activity.

The color changes in wild boar loins are shown in Table 3. The different methods of wild boar meat maturation did not significantly affect the L* parameter of the samples after production (*p* > 0.05). The treatments with the apple vinegar addition (R1 and R2) were characterized by a significantly (*p* < 0.05) lighter color after 14 days of storage. After 28 days of storage, the R1 treatment had a comparable L* value to variant C, and the R2 treatment was significantly (*p* < 0.05) darker. After production, the variants with the addition of vinegar stand out by a significantly (*p* < 0.05) higher proportion of red color (a*) than the control treatment, whereas after 28 days, the trend was reversed. The factor influencing the level of nitrite residues in the product is, among others, the pH value [62]. A lower pH promotes the faster reduction in nitrites to nitric oxide; as the conversion of heme pigments increases, more nitrosomyoglobin is formed, which causes a greater share of a* in the color, which was also observed in our research. Similar significant changes were observed in the results of the b* component; the R1 and R2 treatments were yellower after production and the C variant was more yellow after 28 days of storage (*p* < 0.05). A significant (*p* < 0.05) increase in the a* and b* components in treatment C and R2 during storage was also observed. Whereas, in the R1 variant, an increase in the share of red and yellow color was observed after 14 days, and the decrease in the values of these components occurred after 28 days of storage. Lindahl et al. [63] found that early post mortem temperature, pH, and metabolite levels significantly impacted wild boar meat color, with a higher temperature and lower pH causing a paler, more yellow color due to the inactivation of oxygen-consuming enzymes and protein denaturation. In summary, marinating, storage temperature, post mortem changes, and other factors impact the color, texture, odor, and other sensory properties of wild boar meat. Furthermore, proper handling and processing can help maximize quality and shelf life.

The microbiological quality of fresh or preserved food products is the degree of their safety, microbiological stability, sensory acceptability, and dietary suitability. All tested wild boar loin treatments were characterized by appropriate microbiological quality (Table 4). The initial total viable count (TVC) was not high (6.59–6.91 log CFU/g). After 14 and 28 days of storage, a significant increase in the total number of microorganisms by approximately 1 logarithmic order was observed (7.32–7.61 log CFU/g, *p* < 0.05). The high TVC was mainly due to high lactic acid bacteria (LAB) counts ranging from 5.61 to 7.06 log CFU/g. No *Enterobacteriaceae*, including *Escherichia coli*, were found in all of the tested samples (<1.00 log CFU/g—counts below the detection limit of the plating method). In the control sample (C), coagulase-positive staphylococci of 2.05 and 1.85 log CFU/g were observed at the initial time and after 14 days of storage, respectively. *Staphylococcus aureus* was not found. Because the wild boar loins were macerated by hand, and with raw ripening, unprocessed products may contain trace amounts of staphylococci from production workers. In the research tests with the addition of R1 and R2 fruit vinegar, no *Staphylococcus* bacteria were observed, which proves the antimicrobial properties of the vinegar used. No presence of *Salmonella* spp., *Listeria* spp., and *Campylobacter* spp. was found in any of the tested treatments of loin.

Wild boar meat can lead to food poisoning, as can meat from slaughtered farm animals. Microbiological safety is closely related to the hunting process itself, hygiene, and the appropriate carcass-chilling temperature, as well as the subsequent handling of animal carcasses [9]. Game meat safety requirements in the EU are regulated by EU Regulations 178/2002, 852/2004, 853/2004, and 854/2004 [64,65,66,67]. These laws govern the liability, traceability, and safety of wild game meat, ensuring the same level of control as for meat. In an analysis of the literature by Gomes-Neves et al. [68] and Paulsen et al. [69], special emphasis was placed on checking the presence of *Salmonella* spp., *Yersinia* spp., and *Campylobacter* spp. These microorganisms were characterized as the main potential source of wild boar meat infection. The presence of *Salmonella* spp. and *Campylobacter* spp. bacteria (*Yersinia* spp. not tested) was not found in our research, which indicates the good microbiological quality of the tested raw wild boar loins.

The sensory quality of meat depends on the physical, chemical, and morphological composition of the meat, the ante mortem handling, and the post-slaughter processes combined with the storage method. A comprehensive evaluation is also affected by quality attributes such as taste, smell, color, and individual consumer preferences. Considering consumers’ growing interest in venison, knowledge about the relationship between individual sensory features of wild boar meat seems essential [70]. The sensory evaluation scores of the wild boar meat products were presented in Figure 1a–c.

Immediately after production, all wild boar loins were characterized by a similar odor and flavor profile, with an intense odor and flavor of dried meat (4.40–5.15 c.u.). Control loins C were characterized by a slightly darker, more red color (7.83 c.u.), but this difference was not statistically significant (*p* > 0.05). Samples with the addition of apple vinegar (R1 and R2) were characterized by significantly greater juiciness (6.97–7.13 c.u.) and fragility (4.57–4.69 c.u.) (*p* < 0.05). The loins with the addition of R1 apple vinegar were characterized by significantly the highest overall quality (7.36 c.u.). After 14 days of storage, an intense fatty (3.83 c.u.) and sour (4.81 c.u.) odor was observed in the R2 loins, but these differences were not statistically significant (*p* > 0.05). The loins with the addition of apple vinegar R1 were characterized by significantly greater juiciness (7.01 c.u.) and fragility (4.71 c.u.), as well as overall quality (7.61 c.u.) (*p* < 0.05) compared with the loins C and R2. In turn, after 28 days of storage, experts assessed that the R2 loins were characterized by greater odor intensity (fatty, seasoning, sour, and other odors) (*p* > 0.05) and significantly greater juiciness and fragility (*p* < 0.05). However, in the assessment of overall quality, the significantly highest scores were obtained in the case of R1 loins (7.76 c.u.). The reason for the more distinct smell of the R2 loins may be the method of preparing the loins for the ripening process. The addition of glucose before adding vinegar in the R1 treatment probably softened the characteristic gamey taste, which was reflected in the significantly higher overall quality scores throughout the storage period. The glucose was probably metabolized by the acetic bacteria present in apple vinegar which could have a positive effect on the flavor and aroma profile. Żochowska-Kujawska et al. [71] found that marinating wild boar meat reduces fiber size and connective tissue thickness, resulting in increased fragility and juiciness, which was also observed in our studies.

## 4. Conclusions

There is growing consumer interest in wild boar products, mainly due to their taste and nutritional value. The technology for producing raw-ripened wild boar loins with the addition of apple vinegar developed by the authors may contribute to increasing the consumption and popularity of wild boar meat products. On the one hand, it will facilitate technological processes, as game meat is difficult to process. But the technique of marinating in apple vinegar is simple and can certainly also be used at home. On the other hand, given the current problem of game management due to ASF, it will make it easier for producers to process these products. The use of natural fermentation apple vinegar in the processing technology of wild boar meat products had a positive effect on the sensory properties, especially the juiciness and fragility of the finished product. In terms of overall quality, wild boar loin with apple vinegar is rated better by experts. The desired sensory and physicochemical properties, including the desired color, of the manufactured products were stable during refrigerated storage. It is also important to confirm the bacteriostatic properties of apple vinegar obtained from natural fermentation and the validity of its use in the processing of wild boar meat, which is particularly important from the point of view of the safety and microbiological quality of game products, especially to those that are not subjected to heat treatment.

## Figures and Tables

**Figure 1 foods-12-03975-f001:**
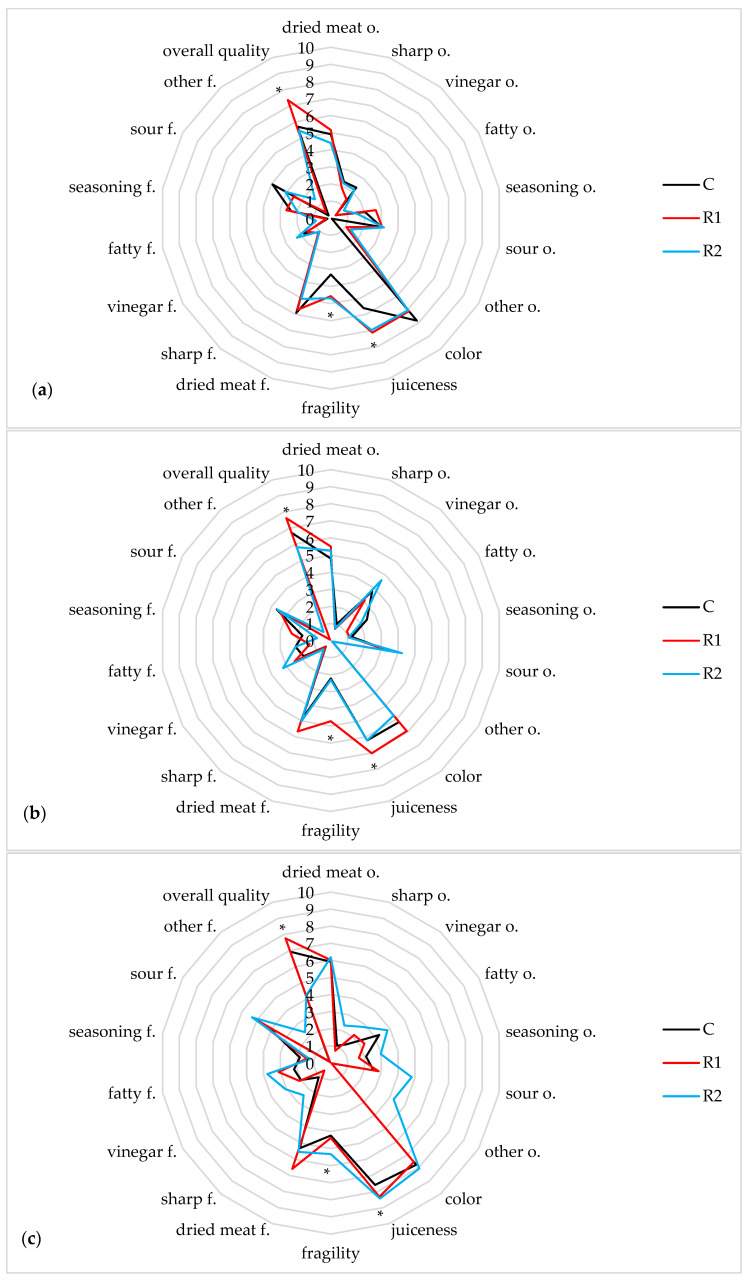
Quality evaluation of wild boar loins according to QDA (**a**) after production, (**b**) after 14 days, and (**c**) after 28 days of refrigerated storage. The values are expressed as means. * statistically significant differences between treatments were observed (*p* < 0.05).

**Table 1 foods-12-03975-t001:** Fatty acid composition and cholesterol content of the tested treatments of raw-ripened wild boar loins.

Parameter	Treatment	Storage (Days)	
		0	28
SFA (%)	C	36.27 ± 3.43 ^bAB^	33.60 ± 1.04 ^aB^
	R1	34.23 ± 1.98 ^aA^	36.32 ± 0.31 ^bAB^
	R2	38.67 ± 1.32 ^bB^	31.07 ± 3.87 ^aA^
MUFA (%)	C	33.88 ± 2.75 ^aAB^	38.62 ± 8.23 ^aB^
	R1	30.92 ± 11.59 ^aA^	43.37 ± 4.72 ^bB^
	R2	42.75 ± 2.09 ^bB^	27.40 ± 5.46 ^aA^
PUFA (%)	C	29.82 ± 5.94 ^aA^	27.72 ± 7.44 ^aAB^
	R1	34.78 ± 12.84 ^bA^	20.18 ± 4.60 ^aA^
	R2	25.70 ± 7.29 ^aA^	31.40 ± 1.54 ^bB^
n-3 (%)	C	5.78 ± 2.67 ^aAB^	7.43 ± 4.90 ^aAB^
	R1	10.15 ± 5.92 ^bB^	4.02 ± 1.41 ^aA^
	R2	3.80 ± 2.32 ^aA^	11.17 ± 4.02 ^bB^
n-6 (%)	C	23.40 ± 3.17 ^aA^	19.68 ± 4.77 ^aA^
	R1	24.05 ± 6.85 ^bA^	15.60 ± 3.27 ^aA^
	R2	20.92 ± 4.92 ^aA^	19.50 ± 2.20 ^aA^
n-6/n-3	C	4.55 ± 1.41	3.55 ± 1.68
	R1	2.99 ± 1.39	4.16 ± 1.08
	R2	6.77 ± 3.32	2.11 ± 1.22
Trans (%)	C	0.47 ± 0.31 ^aA^	0.35 ± 0.08 ^aA^
	R1	0.52 ± 0.24 ^aA^	0.57 ± 0.19 ^aAB^
	R2	0.30 ± 0.00 ^aA^	0.43 ± 0.08 ^bB^
Cholesterol (mg/100 g of product)	C	75.13 ± 10.71 ^aA^	85.28 ± 3.96 ^aB^
	R1	83.08 ± 10.46 ^aA^	83.75 ± 4.94 ^aB^
	R2	80.65 ± 7.11 ^aA^	76.18 ± 2.30 ^aA^

SFA—the sum of the saturated fatty acids; MUFA—the sum of the monounsaturated fatty acids; PUFA—the sum of the polyunsaturated fatty acids; n-3—the sum of the n-3 fatty acids; n-6—the sum of the n-6 fatty acids; Trans—the sum of the trans fatty acids; C—control sample, wild boar loin produced without the addition of apple vinegar, R1—wild boar loin produced with apple vinegar (glucose added with apple vinegar), R2—wild boar tenderloin produced with apple vinegar (glucose added after the marinating process). The values are expressed as means ± SD. Means in the same column followed by different uppercase letters (^A,B^) are significantly different in the time (*p* < 0.05), means in the same row followed by different lowercase letters (^a,b^) are significantly different in the treatment (*p* < 0.05).

**Table 2 foods-12-03975-t002:** pH value, oxidation-reduction potential (ORP), and TBARS index of the tested treatments of raw-ripened wild boar loins.

Parameter	Treatment	Storage (Days)		
		0	14	28
	C	5.50 ± 0.17 ^abB^	5.70 ± 0.04 ^bC^	5.35 ± 0.20 ^aB^
pH	R1	5.36 ± 0.13 ^cAB^	5.24 ± 0.02 ^bA^	5.10 ± 0.04 ^aA^
	R2	5.17 ± 0.19 ^aA^	5.33 ± 0.01 ^bB^	5.34 ± 0.08 ^bB^
	C	420.82 ± 4.02 ^aB^	475.32 ± 2.75 ^bA^	498.57 ± 17.42 ^cA^
ORP (mV)	R1	421.83 ± 6.08 ^aB^	488.48 ± 0.64 ^bB^	498.42 ± 1.33 ^cA^
	R2	409.75 ± 4.31 ^aA^	487.40 ± 0.68 ^bB^	486.77 ± 5.15 ^bA^
TBARS index (mg MDA/kg of product)	C	0.878 ± 0.067 ^bB^	0.765 ± 0.096 ^bB^	0.593 ± 0.100 ^aA^
	R1	1.013 ± 0.171 ^bB^	0.461 ± 0.009 ^aA^	0.575 ± 0.070 ^aA^
	R2	0.545 ± 0.074 ^aA^	0.822 ± 0.077 ^bB^	1.294 ± 0.061 ^cB^

C—control sample, wild boar loin produced without the addition of apple vinegar, R1—wild boar loin produced with apple vinegar (glucose added with apple vinegar), R2—wild boar tenderloin produced with apple vinegar (glucose added after the marinating process). The values are expressed as means ± SD. Means in the same column followed by different uppercase letters (^A–C^) are significantly different in the time (*p* < 0.05); means in the same row followed by different lowercase letters (^a–c^) are significantly different in the treatment (*p* < 0.05).

**Table 3 foods-12-03975-t003:** Color parameters of the tested treatments of raw-ripened wild boar loins.

Parameter	Treatment	Storage (Days)		
		0	14	28
L*	C	40.00 ± 2.07 ^aB^	32.39 ± 8.58 ^aA^	41.55 ± 2.81 ^bB^
	R1	39.53 ± 2.15 ^aB^	36.50 ± 1.55 ^bA^	40.65 ± 2.34 ^bB^
	R2	39.67 ± 2.36 ^aB^	40.64 ± 1.46 ^cB^	38.25 ± 1.74 ^aA^
a*	C	7.45 ± 1.65 ^aA^	14.27 ± 3.63 ^bB^	14.10 ± 1.41 ^bB^
	R1	9.29 ± 2.06 ^bA^	15.60 ± 2.06 ^bC^	12.23 ± 2.41 ^aB^
	R2	10.00 ± 2.60 ^bA^	10.71 ± 1.91 ^aA^	12.81 ± 3.29 ^abB^
b*	C	5.01 ± 0.82 ^aA^	7.18 ± 1.51 ^aB^	8.89 ± 1.17 ^bC^
	R1	6.09 ± 0.95 ^bA^	7.83 ± 0.70 ^bC^	7.08 ± 0.85 ^aB^
	R2	6.15 ± 0.95 ^bA^	6.88 ± 0.67 ^aB^	7.10 ± 1.02 ^aB^

L* (lightness), a* (redness: green to red), b* (yellowness: blue to yellow); C—control sample, wild boar loin produced without the addition of apple vinegar, R1—wild boar loin produced with apple vinegar (glucose added with apple vinegar), R2—wild boar tenderloin produced with apple vinegar (glucose added after the marinating process). The values are expressed as means ± SD. Means in the same column followed by different uppercase letters (^A–C^) are significantly different in the time (*p* < 0.05); means in the same row followed by different lowercase letters (^a–c^) are significantly different in the treatment (*p* < 0.05).

**Table 4 foods-12-03975-t004:** Microbiological analysis of the tested treatments of raw-ripened wild boar loins.

Parameter	Treatment	Storage (Days)		
		0	14	28
TVC (log CFU/g)	C	6.59 ± 0.13 ^aA^	7.42 ± 0.29 ^aB^	7.48 ± 0.34 ^aB^
	R1	6.91 ± 0.11 ^aA^	7.32 ± 0.13 ^aB^	7.49 ± 0.03 ^aB^
	R2	6.59 ± 0.08 ^aA^	7.48 ± 0.14 ^aB^	7.61 ± 0.04 ^aB^
LAB (log CFU/g)	C	6.79 ± 0.11 ^bA^	6.87 ± 0.01 ^aA^	7.06 ± 0.02 ^bA^
	R1	6.41 ± 0.52 ^bA^	6.05 ± 0.05 ^aA^	6.36 ± 0.08 ^aA^
	R2	5.61 ± 0.10 ^aA^	6.19 ± 0.18 ^aA^	6.18 ± 0.17 ^aA^
ENT (log CFU/g)	C	<1.00 ^aA^	<1.00 ^aA^	<1.00 ^aA^
	R1	<1.00 ^aA^	<1.00 ^aA^	<1.00 ^aA^
	R2	<1.00 ^aA^	<1.00 ^aA^	<1.00 ^aA^
EC (log CFU/g)	C	<1.00 ^aA^	<1.00 ^aA^	<1.00 ^aA^
	R1	<1.00 ^aA^	<1.00 ^aA^	<1.00 ^aA^
	R2	<1.00 ^aA^	<1.00 ^aA^	<1.00 ^aA^
SA (log CFU/g)	C	2.05 ± 0.08 ^bB^	1.85 ± 0.09 ^bB^	<1.00 ^aA^
	R1	<1.00 ^aA^	<1.00 ^aA^	<1.00 ^aA^
	R2	<1.00 ^aA^	<1.00 ^aA^	<1.00 ^aA^
SAL	C	nd	nd	nd
	R1	nd	nd	nd
	R2	nd	nd	nd
LM	C	nd	nd	nd
	R1	nd	nd	nd
	R2	nd	nd	nd
	C	nd	nd	nd
CAMP	R1	nd	nd	nd
	R2	nd	nd	nd

TVC—total viable count; LAB—lactic acid bacteria; ENT—bacteria *Enterobacteriaceae*; EC—*Escherichia coli*; SA—coagulase-positive staphylococci (*Staphylococcus aureus* and other species); SAL—*Salmonella* spp.; LM—*Listeria* spp. including *Listeria monocytogenes*; CAMP—*Campylobacter* spp.; <1.00—counts below the detection limit of the plating method; nd—not detected; C—control sample, wild boar loin produced without the addition of apple vinegar, R1—wild boar loin produced with apple vinegar (glucose added with apple vinegar), R2—wild boar tenderloin produced with apple vinegar (glucose added after the marinating process). The values are expressed as means ± SD. Results in the same column followed by different uppercase letters (^A,B^) are significantly different in the time (*p* < 0.05); means in the same row followed by different lowercase letters (^a,b^) are significantly different in the treatment (*p* < 0.05).

## Data Availability

The data used to support the findings of this study can be made available by the corresponding author upon request.

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
