# Peer review of "The Use of Apple Vinegar from Natural Fermentation in the Technology Production of Raw-Ripened Wild Boar Loins"

_foods, 2023, doi:10.3390/foods12213975_

Round 1
Reviewer 1 Report
Comments and Suggestions for Authors
This study developed a method to product the wild boar loin using apple vinegar and detected the impact of apple vinegar on the microbiological, sensory quality, and physico-chemical parameters of the product. The study may provide important reference for the process and promotion of wild boar meat products. However, some clarification and improvements should be made before acceptance.
1. The “introduction” section provides an excess of information about the consumption of wild boar, rather than the information about the production or process of the meat, which is not consistent with the aim of this study. Authors should provide more information related to the conventional processing methods of wild boar meat, as well as information related to the application of apple vinegar in food processing.
2. Commonly, wild animals only live in sparsely populated areas or specific hunting areas, so how do the authors think about the impact of wild animal consumption on biodiversity? Or the authors intend to feed the domestic pigs in the way as wild animals? So, what do the authors think about the productivity of this kind of feeding?
3. Authors should provide the chemical composition of 4% apple vinegar, as well as an explanation of why the 4% concentration was chosen over other concentrations.
4. As shown in table1, different treatments of wild meat changed the fatty acid composition and cholesterol content. So, can the authors explain the mechanism by which this change occurs? For example, the PUFA and n-3 FA was significantly increased on day0 after R1treatment, while reduced to the lowest on Day28. How this happened?
5. Multiple indicators of the sensory quality were detected and no significant difference was found in most of the indicators. So, is it appropriate to conclude that the use of apple vinegar has a positive effect on the sensory properties, especially the juiciness and tenderness of the finished product?
6. The 1st paragraph of conclusion section seems to be information about background, rather than the result or discussion about this study.
Author Response
Replies to review 1
We would like to thank the reviewer for his valid comments regarding our manuscript. We have taken into account all the Reviewer's comments. We made the corrections in the Manuscript in the "Track Changes" function. We hope that our corrections will meet the Reviewer's requirements.
Question 1. The “introduction” section provides an excess of information about the consumption of wild boar, rather than the information about the production or process of the meat, which is not consistent with the aim of this study. Authors should provide more information related to the conventional processing methods of wild boar meat, as well as information related to the application of apple vinegar in food processing.
Answer: Lines 82-96, 101-106. The Introduction section includes information on how to process wild boar meat (marinating) and information on apple vinegar as a marinade ingredient.
Question 2. Commonly, wild animals only live in sparsely populated areas or specific hunting areas, so how do the authors think about the impact of wild animal consumption on biodiversity? Or the authors intend to feed the domestic pigs in the way as wild animals? So, what do the authors think about the productivity of this kind of feeding?
Answer: Thank you for this very interesting question. We believe that these specific hunting areas, where there is little human interference in the environment, are extremely important and interesting. Wild animals have a huge impact on the preservation of biodiversity in nature. Biodiversity is the foundation of life. It is essential both for people and for environmental and climate protection. It provides people with food, drinking water and clean air and plays an important role in maintaining the balance of nature. In the era of total industrialization, expansion of cities, as well as diet-related diseases in people, we should strive to preserve naturalness in our lives as much as possible. One of the ways to protect biodiversity is the protection of plant, animal and fungi species, known to all of us. Another is the creation of nature protection areas such as national parks, nature reserves, landscape parks, protected landscape areas and other surface forms of nature protection.
The authors do not intend to feed domestic pigs as wild animals, due to different market demands. We cannot imagine it, after all, wild animals occur naturally in the environment and choose their own food source. We are in favor of enriching our diet with nutrient-rich game meat, but at the moment we are unable to change people's mentality regarding eating meat from industrially produced slaughter animals.
Question 3. Authors should provide the chemical composition of 4% apple vinegar, as well as an explanation of why the 4% concentration was chosen over other concentrations.
Answer: Lines 129-145. An extensive description of the apple vinegar used was added.
The research results presented in this manuscript are part of the Incubator of Innovation 4.0 project (detailed information about the project can be found in the Funding section). In the first stage of the research, the authors prepared pilot wild boar products with various additions of apple vinegar (control sample without apple vinegar, 2, 3, 4 and 5% of apple vinegar added respectively). Based on the physico-chemical, microbiological and sensory tests carried out, a 4% addition of apple vinegar was selected for the production of wild boar loins. This information has been supplemented in the 2.1. Materials section (Lines 148-154).
Question 4. As shown in table1, different treatments of wild meat changed the fatty acid composition and cholesterol content. So, can the authors explain the mechanism by which this change occurs? For example, the PUFA and n-3 FA was significantly increased on day0 after R1treatment, while reduced to the lowest on Day28. How this happened?
Answer: The results of the fatty acid composition and cholesterol content caused many problems in interpretation. The fatty acid composition of meat (both domestic and wild animals) is influenced by nutrition, including the type and fat content of food (Wood & Enser, 1997). Genetic factors also influence the composition. The content of saturated (SFA) and monounsaturated (MUFA) fatty acids increases faster with increasing body fat. Differences in fatty acid composition between breeds and genotypes can be largely explained by differences in adiposity (De Smet et al., 2004; Wood et al., 2004).
In our research, each time we took a new loin for testing. These loins came from various wild boars because they were purchased and produced industrially. We would not be able to collect identical loins from specific wild boars because we would lack material for testing. Due to the fact that we took a new loin for each study, they could differ in fatty acid composition and cholesterol content because they came from different individuals. These individuals could lead a different lifestyle, eat different food at a given time and live in a different part of the environment.
In response to the reviewer's comment, we have supplemented the information on possible changes in the composition of fatty acids (Lines 262-302).
De Smet, S., Raes, K., & Demeyer, D. (2004). Meat fatty acid composition as affected by fatness and genetic factors: a review. Animal Research, 53(2), 81-98.
Wood, J. D., & Enser, M. (1997). Factors influencing fatty acids in meat and the role of antioxidants in improving meat quality. British journal of Nutrition, 78(1), S49-S60.
Wood, J. D., Richardson, R. I., Nute, G. R., Fisher, A. V., Campo, M. M., Kasapidou, E., ... & Enser, M. (2004). Effects of fatty acids on meat quality: a review. Meat science, 66(1), 21-32.
Question 5. Multiple indicators of the sensory quality were detected and no significant difference was found in most of the indicators. So, is it appropriate to conclude that the use of apple vinegar has a positive effect on the sensory properties, especially the juiciness and tenderness of the finished product?
Answer: Lines 469-520. Thank you for your valid attention. We have corrected the sensory evaluation chart, where we correctly placed statistically significant results. We have also prepared a new description of sensory evaluation.
Question 6. The 1st paragraph of conclusion section seems to be information about background, rather than the result or discussion about this study.
Answer: Lines 522-546. Thank you for your valid attention. We have modified the Conclusions as suggested by the reviewer.

Reviewer 2 Report
Comments and Suggestions for Authors
Manuscript ID: sustainability-2665498
Title: The use of apple vinegar from natural fermentation in the technology
of raw-ripened wild boar loin production
Authors: Anna Łepecka *, Piotr Szymański, Anna Okoń, Beata Łaszkiewicz,
Sylwia Onacik-Gür, Dorota Zielińska, Zbigniew J. Dolatowski
The research aimed to develop a technology for the production of wild boar loin using apple vinegar for marinating and to determine the impact of apple vinegar on the microbiological and sensory quality, and physico-chemical parameters of the product. Even though this manuscript is interesting, several concerns need to be addressed before further consideration.
Please see the specific comment in PDF file and reply to my comment point-by-point.

Comments on the Quality of English LanguageAuthor Response
Replies to review 2
We would like to thank the reviewer for his valid comments regarding our manuscript. We have taken into account all the Reviewer's comments. We made the corrections in the Manuscript in the "Track Changes" function. We hope that our corrections will meet the Reviewer's requirements.
Lines 14-31. We have modified the Abstract according to the reviewer's suggestion. We have expanded the abbreviations C, R1, R2, MDA, added a description of the methodology and completed the missing information.
Line 26. "Health safety of products" means health safety, i.e. the absence of microflora that spoils food and pathogens that could cause diseases.
Line 32. We have removed the ASF abbreviation from Keywords and replaced it with "acetic bacteria".
Lines 47-49. We have supplemented the information on the reasons for the low consumption of wild boar meat in Europe.
Line 50. We have corrected the ASF abbreviation as suggested by the Reviewer.
Lines 56-67. We have supplemented information on the mechanisms related to the high content of vitamins and nutrients in the meat of wild animals.
Lines 82-96. The Introduction section has been supplemented with information on the possibilities of processing wild boar meat (marinating) and the information on apple vinegar as an ingredient of meat marinade has been supplemented.
Lines 99-100. IBPRS-PIB abbreviation was explained.
Lines 110-112. According to the reviewer's suggestion, a research hypothesis was added.
Lines 129-145. A detailed description of the basic composition and physico-chemical properties used to produce apple vinegar was presented.
Lines 149-151. More detailed information on the raw material (wild boar meat) has been added.
Lines 152-158. Information on preliminary tests and the reason for choosing 4% apple vinegar added to the marinade has been modified.
Line 212. The title of the subsection "Instrumental measurement of color" has been corrected
Lines 262-266. The sentence has been corrected according to the Reviewer's suggestion.
Line 268. WHO abbreviation added.
Lines 283-289, 297-302. Possible mechanisms causing high cholesterol levels and changes in fatty acid ratios in wild boar loins were added.
Table 1, 2, 3, 4. Missing explanations of abbreviations have been completed.
Lines 326-329. The information has been supplemented in accordance with the Reviewer's suggestion.
Lines 410-411. It was decided to remove the sentence "Genotype and peri-mortem stress also influenced color".
Lines 423-425. We have modified the sentences on microbiological quality.
Lines 469-520. Thank you for your important attention. We have corrected the sensory evaluation chart, where we correctly placed statistics significant results. We have also prepared a new description of sensory evaluation.
Lines 523-546. Thank you for your important attention. We have modified the Conclusions as suggested by the reviewer.

Round 2
Reviewer 2 Report
Comments and Suggestions for Authors
The authors addressed all of the criticisms and provide their recommendation to accept the manuscript in its current form.